# Identification of the potential active site of the septal peptidoglycan polymerase FtsW

Ying Li[1]☯, Adrien Boes[2]☯¤, Yuanyuan Cui[1], Shan Zhao[1], Qingzhen Liao[1], Han Gong[1], Eefjan Breukink[3], Joe Lutkenhaus[4], Mohammed Terrak[2]*, Shishen Du[1]*

1 Department of Microbiology, Hubei Key Laboratory of Cell Homeostasis, College of Life Sciences, Wuhan University, Wuhan, Hubei, China, 2 InBioS-Centre d'Ingénierie des Protéines, Liège University, Liège, Belgium, 3 Membrane Biochemistry and Biophysics, Department of Chemistry, Faculty of Science, Utrecht University, Utrecht, The Netherlands, 4 Department of Microbiology, Molecular Genetics and Immunology, University of Kansas Medical Center, Kansas City, Kansas, United States of America

☯ These authors contributed equally to this work.
¤ Current address: CER Groupe—Immunobiology Laboratory, Novalis Science Park, Aye, Belgium
* mterrak@uliege.be (MT); ssdu@whu.edu.cn (SD)

**Data Availability Statement:** All relevant data are within the manuscript and its Supporting Information files.

**Funding:** This study was supported by National Natural Science Foundation of China (grant

## Abstract

SEDS (Shape, Elongation, Division and Sporulation) proteins are widely conserved peptidoglycan (PG) glycosyltransferases that form complexes with class B penicillin-binding proteins (bPBPs, with transpeptidase activity) to synthesize PG during bacterial cell growth and division. Because of their crucial roles in bacterial morphogenesis, SEDS proteins are one of the most promising targets for the development of new antibiotics. However, how SEDS proteins recognize their substrate lipid II, the building block of the PG layer, and polymerize it into glycan strands is still not clear. In this study, we isolated and characterized dominant-negative alleles of FtsW, a SEDS protein critical for septal PG synthesis during bacterial cytokinesis. Interestingly, most of the dominant-negative FtsW mutations reside in extracellular loops that are highly conserved in the SEDS family. Moreover, these mutations are scattered around a central cavity in a modeled FtsW structure, which has been proposed to be the active site of SEDS proteins. Consistent with this, we found that these mutations blocked septal PG synthesis but did not affect FtsW localization to the division site, interaction with its partners nor its substrate lipid II. Taken together, these results suggest that the residues corresponding to the dominant-negative mutations likely constitute the active site of FtsW, which may aid in the design of FtsW inhibitors.

## Author summary

SEDS (Shape, Elongation, Division and Sporulation) proteins are widely conserved peptidoglycan polymerases that play critical roles in cell elongation and cell division in rod-shaped bacteria. However, how they catalyze PG polymerization remains poorly understood. In this study, we isolated and characterized a set of dominant-negative mutations in the SEDS protein FtsW, which synthesizes septal peptidoglycan during cell division in most bacteria. Our results revealed that the dominant-negative mutations disrupt FtsW's ability to synthesize peptidoglycan, but do not affect its other activities, suggesting that the corresponding amino acids may constitute the active site of FtsW.

32070032, http://www.nsfc.gov.cn/), the
Fundamental Research Funds for the Central
Universities (grant 2042021kf0198) and Wuhan
University (https://www.whu.edu.cn/) to S.D.;
National Institute of General Medical Sciences
(grant GM029764; https://www.nigms.nih.gov/) to
J.L.; the Fonds de la Recherche Scientifique
(FRS_FNRS), (grant CDR J.0030.18) to M.T.; and
Fonds pour la Formation à la Recherche dans
l'Industrie et dans l'Agriculture (fellowship 1.
E.038.17) to A.B. The funders had no role in study
design, data collection and analysis, decision to
publish, or preparation of the manuscript.

**Competing interests:** The authors have declared
that no competing interests exist.

## Introduction

The cytoplasmic membrane of most bacteria is surrounded by a mesh-like peptidoglycan (PG) layer that is composed of glycan strands of alternating N-acetylglucosamine (GlcNAc) and N-acetylmuramic acid (MurNAc) residues crosslinked by a peptide bridge [1,2]. The PG layer not only determines the shape of the bacterial cell, but also protects it from osmotic rupture. Therefore, the synthesis of the PG layer and maintenance of its integrity are crucial for the survival of bacterial cells. PG is synthesized from its precursor lipid II, which is polymerized and cross-linked into the PG matrix by glycosyltransferase (PGTase) and transpeptidase (TPase) enzymes, respectively [1,2].

Two types of PG synthases have been discovered. The first is the class A penicillin binding proteins (aPBPs), which contain both a PGTase domain and a TPase domain [3,4]. Thus, aPBPs can catalyze both the polymerization and cross-linking reactions needed for PG synthesis. The second type of PG synthase consists of a complex between a SEDS (Shape, Elongation, Division and Sporulation) protein and a class B PBP (bPBP) [5–11]. SEDS proteins are highly conserved multipass transmembrane proteins which were originally thought to function as a lipid II flippase [12] but were recently demonstrated to have PGTase activity [5,6], whereas bPBPs are monofunctional TPases [3]. Together, they polymerize and cross-link PG in a coordinated manner. For decades, aPBPs were thought to be the primary PG synthases in bacteria [13]. However, in the last few years accumulating evidence indicates that SEDS-bPBP complexes are responsible for synthesizing PG during bacterial cell growth and division [5–11], while aPBPs appear to be largely involved in repair and maintenance of the PG layer [7,14–18]. Nonetheless, both sets of PG synthases are essential for the survival of most bacteria having a cell wall, although some manage to survive without aPBPs under certain circumstances [5,19–22].

Most rod-shaped bacteria employ two distinct multi-protein complexes organized by cytoskeletal filaments to direct the synthesis of PG during growth and division [1]. The elongasome or Rod complex promotes cell elongation and is organized by filaments of the actin-like MreB protein [23]. This complex contains a SEDS-bPBP pair formed by RodA and PBP2 to insert new glycan strands into the PG layer perpendicular to the long axis of the cell [5,10,11,23–26]. The divisome (septal ring) utilizes a SEDS-bPBP pair formed by FtsW and FtsI (PBP3) to synthesize septal PG during cytokinesis and is organized by the Z ring formed by filaments of the tubulin-like FtsZ protein [8,27–29]. Understanding how the activities of RodA-PBP2 and FtsW-FtsI are regulated is critical for elucidating the mechanisms controlling bacterial cell elongation and cell division.

Accumulated evidence suggests that the activity of the SEDS-bPBP complexes *in vivo* is regulated by an activation pathway that goes through bPBP to stimulate the PG polymerase activity of its cognate SEDS protein [10,11,30–32]. In the elongasome, the activity of RodA-PBP2 is believed to be regulated by MreC and MreD [11,33], two critical components of the elongasome. Mutations in MreC, which block the activity of the elongasome, can be suppressed by activating mutations in RodA or PBP2 that increase the activity of the RodA-PBP2 complex *in vivo* and *in vitro* and appear to bypass the activation step [11]. Consistent with this, MreC from *Helicobacter pylori* has been shown to interact and co-crystalize with PBP2 and disruption of this interaction blocks the activity of the elongasome [34]. Thus, the model emerging from these studies is that MreC (influenced by MreD) stimulates the activity of RodA-PBP2 by causing a conformational change in PBP2 which ultimately leads to the activation of RodA's polymerase activity [11]. Structural analysis of a RodA-PBP2 complex from *Thermus thermophilus* suggests that an interaction between the pedestal domain of PBP2 and the 4th extracellular loop (ECL4) of RodA is critical for the stimulation [10]. In agreement with this, mutations

in the pedestal domain of PBP2 or in the ECL4 of RodA that disrupt this interaction dramatically reduce the activation of RodA [10], whereas a different set of mutations in the same region increase the activity of RodA *in vitro* and promote the activity of the elongasome *in vivo* [11].

In the divisome, an activation pathway also governs the activity of the FtsW-FtsI complex. A number of cell division proteins or protein complexes, including FtsN, FtsA, FtsEX and FtsQLB, are involved in this regulation [35–37]. FtsN is believed to be the trigger for septal PG synthesis mediated by FtsW-FtsI [38,39]. Activating mutations in FtsA, FtsB or FtsL can bypass FtsN for cell division, suggesting that FtsN acts by switching FtsA in the cytoplasm and FtsQLB in the periplasm to active forms that stimulate the activity of FtsW-FtsI [35,36]. Consistent with this model, purified FtsQLB activates the polymerase activity of the FtsW-FtsI complex *in vitro* with FtsL playing a key role [31], likely through an interaction with FtsI [30]. Moreover, activating mutations in FtsW or FtsI can bypass the entire activation pathway initiated by FtsN [32]. Strikingly, most of the activating mutations occur in the pedestal domain of FtsI or the ELC4 of FtsW [32,40,41]. On the other hand, other mutations in these regions block the activity of FtsW-FtsI, presumably by disrupting the interaction [30,32]. Thus, analogous to RodA activation, the polymerase activity of FtsW is stimulated by FtsI as a result of an activation pathway that modulates the interaction of the pedestal domain of the bPBP and the ECL4 of the SEDS protein.

Despite the recent advance in understanding the regulation of SEDS-bPBPs' activity i*n vivo*, how SEDS proteins recognize their substrate and polymerize it into glycan strands remains poorly understood. Although SEDS proteins are PGTases, they have no homology to the PGTase domain of class A PBPs and have only weak overall similarity to O-antigen ligases, but no conserved sequence motifs with it or other glycosyltransferases [4]. The structure of RodA contains a long hydrophobic groove between the second and third transmembrane (TM) domains [9]. This groove and a number of highly conserved residues adjacent to it may be the binding site for lipid II. The structure of RodA also reveals a large central cavity facing the periplasmic face of the protein, which contains highly conserved residues in the ECLs [9]. Mutational analysis of several residues in this cavity showed that they are critical for RodA function in both *E. coli* and *B. subtilis* [9], suggesting that this cavity may be the active site of RodA and other SEDS proteins. Interestingly, another membrane accessible cavity was found in RodA in the structure of the RodA-PBP2 complex, which is occluded by TM7 in the structure of RodA alone [10]. Since this cavity is large enough to bind a lipid II molecule [10], it was suggested that this cavity is a substrate entry or exit site for lipid II. However, structures of RodA with lipid II bound have not been obtained yet, so whether the above predictions are true is currently unknown.

In this study, we investigated the function of FtsW by isolating dominant-negative alleles using a wrinkled-colony-based screen. Characterization of these mutations revealed that they block septal PG synthesis without affecting other activities of FtsW, suggesting that the mutations disrupt the active site of FtsW. Since these mutations are scattered around the central cavity in an FtsW structure predicted by AlphaFold2, it provides additional evidence that the central cavity of SEDS proteins is the active site for their polymerase activity.

## Results

### Winkled-colony-based screen for dominant-negative FtsW mutants

During bacterial cytokinesis, FtsW interacts with other division proteins including FtsI to form the divisome to synthesize septal PG. However, how FtsW interacts with other division proteins and synthesizes PG is incompletely understood. Non-functional FtsW variants are

key to dissect these interactions and to reveal the enzymatic mechanism, but so far few inactive FtsW mutants have been reported and characterized. Screening for nonfunctional FtsW mutants by a complementation test would be inefficient as mutants that are either unstable or poorly expressed would have to be eliminated. To avoid these problems, we screened for dominant-negative FtsW mutants, since they should be stable and form complexes with its binding partners, but lack some essential function. To do this, we took advantage of the wrinkled-colony morphology displayed by *E. coli* colonies containing filamentous cells due to a cell division defect. This approach has been used successfully to isolate FtsQ and FtsL mutants with reduced function [30,42]. We reasoned that ectopic expression of a dominant-negative FtsW variant in a wild-type strain at an appropriate level would cause a partial inhibition of cell division, giving rise to wrinkled-colonies, which could be easily identified by visual inspection.

As a proof of principle, we tested two inactive mutants of FtsW to see if they generated wrinkled-colonies. One has an alanine substitution of the putative catalytic residue of FtsW (D297). The corresponding mutation in FtsW from *Pseudomonas aeruginosa* and *Staphylococcus aureus* has been shown to be non-functional and dominant-negative [8]. The other is M269K, which is defective in the activation step *in vivo* as determined previously [32]. As shown in Fig 1A, overexpression of either mutant in a wild type strain blocked colony formation, indicating that they were dominant-negative over the wild type FtsW protein. Notably, the catalytic D297A mutant displayed a much greater toxicity than the M269K mutant (Fig 1A), presumably because the D297A mutant was completely inactive whereas the M269K mutant retained basal activity. As expected, cells overexpressing wild type FtsW yielded rounded and smooth colonies on a plate with 30 μM IPTG to induce its expression, whereas cells overexpressing FtsW$^{D297A}$ produced flat and wrinkled colonies (Fig 1B). Examination of the cells from these colonies confirmed that the wrinkled-colony morphology was due to a partial inhibition of cell division. Thus, inactive FtsW mutants were dominant-negative and modest overexpression produced a wrinkled-colony phenotype, which can be exploited to isolate additional dominant-negative mutants.

To screen for dominant-negative FtsW mutants, we created a plasmid library carrying a PCR-mutagenized copy of FtsW under the control of an IPTG-inducible promoter and transformed it into a wild-type strain JS238. Based upon our results with the M269K and D297A mutants, transformants were plated on LB plates containing 30 μM IPTG. As expected, the majority of the colonies were rounded and smooth, likely because they were formed by cells expressing wild-type FtsW or non-toxic mutants. However, about 3 to 5% of the transformants displayed a wrinkled-colony morphology at this IPTG concentration. Microscopic analysis of the cells from these wrinkled-colonies revealed that they were chaining and filamentous (Fig 2A), confirming that cell division was partially inhibited. 25 wrinkled-colonies were randomly picked and restreaked on LB plates with or without IPTG to confirm their phenotype. Plasmids were then isolated from these wrinkled-colonies and transformed back into the parental strain to confirm that the phenotype was linked to the plasmid. Of the twenty-five plasmids isolated, twenty-two reproduced the wrinkled-colony phenotype on LB plates with IPTG. As a result, the *ftsW* coding sequence from these plasmids was sequenced to determine the causative mutations.

## Amino acid substitutions in the extracellular loops of FtsW result in dominant-negative mutants

Sequencing FtsW from the 22 plasmids that produced wrinkled-colonies when ectopically expressed in a wild type strain yielded a total of 20 dominant-negative *ftsW* alleles. Ten of these alleles contained a single amino acid substitution in FtsW and six alleles encoded FtsW

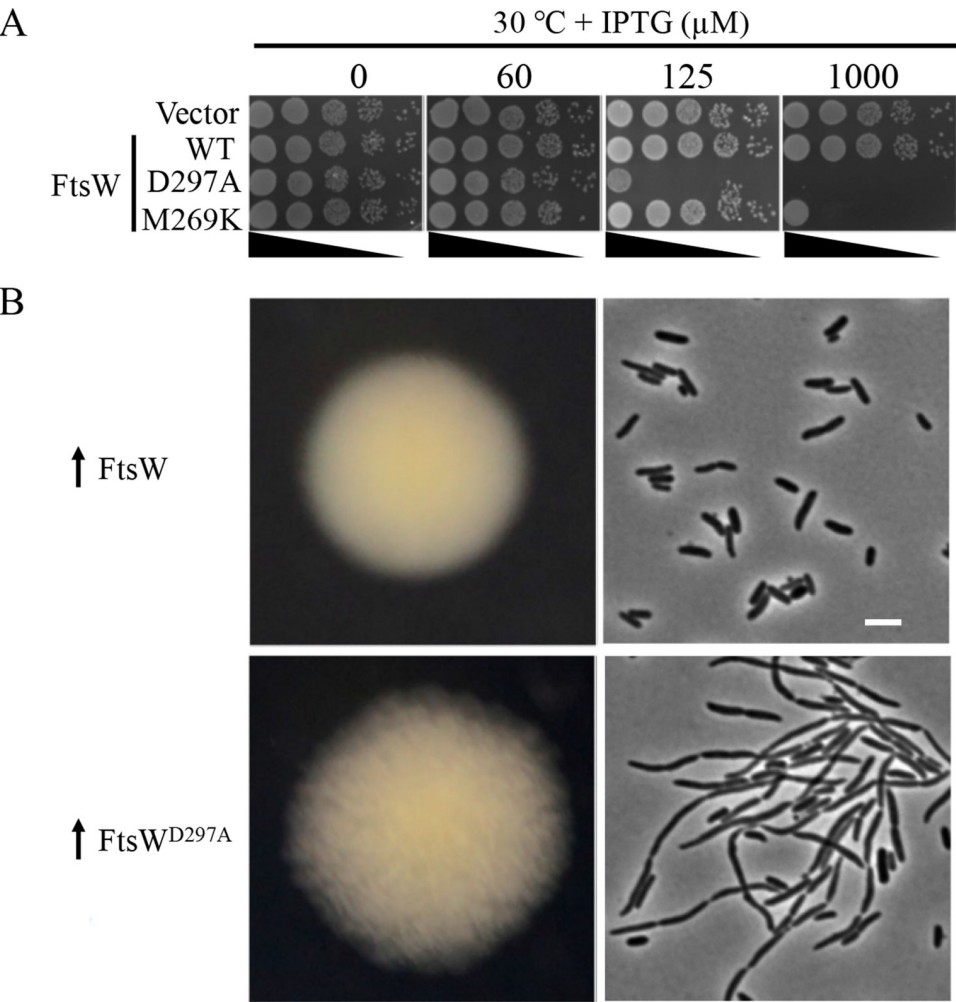

**Fig 1. Modest overexpression of non-functional FtsW mutants are dominant-negative and produce wrinkled-colonies.** (A). A spot test of the dominant-negative effect of inactive FtsW mutants. Plasmids pDSW208, pSEB429 (pDSW208, $P_{204}$::*ftsW*) or its derivatives harboring either the putative catalytic mutation (D297A) or an activation defective mutation (M269K) were transformed into strain JS238 on LB plates with ampicillin and glucose. The next day, a single transformant of each resulting strain was resuspended in 1 ml of LB medium, serially diluted and 3 μl of each dilution was spotted on LB plates with antibiotics, with or without increasing concentrations of IPTG. Plates were incubated at 30°C overnight and photographed. (B) Colony and cell morphology of *E. coli* cells modestly overexpressing FtsW^D297A. Plasmids expressing wild type FtsW or FtsW^D297A were transformed into JS238 as described in (A), but plated on LB plates with 30 μM IPTG. Cells within the colonies were picked and examined. Scale bar, 3 μm.

proteins with a single amino acid substitution and additional silent mutations. The remaining four produced FtsW proteins with two or three amino acids changes, but subsequent analysis revealed that the dominant-negative effect was mainly due to only one of the substitutions. A number of these alleles were isolated more than once or with different substitutions at the same residues (Table 1), such as A135T (twice) and R243H/L/P/S. A glutamine substitution of R243 in *E. coli* FtsW has also been shown to be lethal and dominant-negative in a previous study [43]. Notably, many of them displayed greater toxicity than the putative catalytic mutant D297A and all displayed greater toxicity than the activation mutant (M269K) (Figs 1A and 2B). In total, 18 dominant-negative mutations affecting 12 residues of FtsW were identified (Table 1). As expected, none of these dominant-negative FtsW variants complemented an

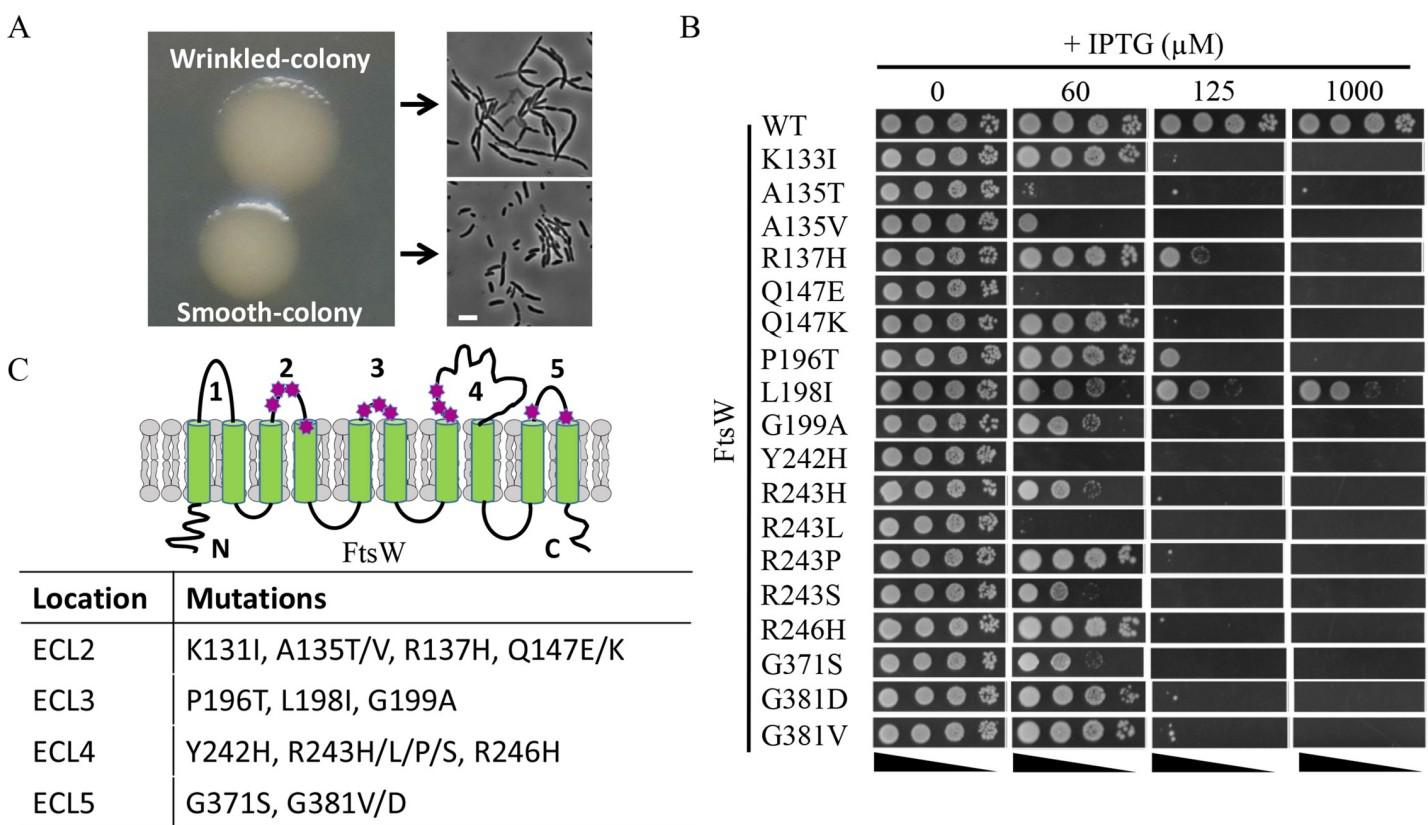

**Fig 2. Isolation of dominant-negative FtsW mutants.** (A) Wrinkled-colony-based screen for dominant-negative FtsW mutants. A plasmid library harboring mutagenized *ftsW* expressed from an IPTG-inducible promoter was transformed into strain JS238 and transformants selected on LB plates with ampicillin and 30 μM IPTG. Most colonies were rounded and smooth, but a small percentage of colonies displayed a flat and wrinkled morphology. Inspection of the two types of colonies revealed that the wrinkled colonies consisted of chaining and filamentous cells. Scale bar, 3 μm (B) Toxicity of dominant-negative FtsW mutants. Plasmids harboring the indicated alleles of *ftsW* were transformed into JS238 and transformants selected on LB plates with ampicillin and glucose. The spot test was performed as in Fig 1A. (C) Location of dominant-negative *ftsW* mutations in a topology model of FtsW. These mutations, indicated by a magenta heptagon, were clustered in ECL2, ECL3, ECL4 and ECL5 of FtsW.

*ftsW* depletion strain (S1 Fig), indicating that they were defective in some essential FtsW activity. Based upon the topology of FtsW determined in a previous study [44], all substitutions were located in the extracellular loops (ECLs) or in TMs abutting the ECLs (6 in ECL2, 3 in ECL3, 6 in ECL4 and 3 in ECL5) (Fig 2C). Alignment of FtsW and its paralog RodA revealed that most of the mutated residues are highly conserved in the SEDS protein family (S2 Fig), including Q147 in ELC2, P196 in ELC3, R246 in ELC4 and G371 and G381 in ECL5. Several of the corresponding residues in *B. subtilis* RodA were shown to be immutable or with limited mutability in a previous study [5](S3 Fig). Moreover, when these substitutions were mapped to a model of *E. coli* FtsW generated by AlphaFold2 [45], the corresponding residues appeared to be scattered around a cavity facing the periplasm (Fig 3A and 3B). Based on structural and mutational analysis of *B. subtilis* RodA, it was suggested that this cavity was the active site of SEDS proteins [9]. Interestingly, residues critical for the function of *B. subtilis* RodA, which were determined by mutagenesis followed by high-throughput sequencing (Mutseq) [5], are also scattered around the cavity of RodA when mapped to a model of the *B. subtilis* RodA structure (S3 Fig). Thus, our screen for dominant-negative FtsW mutants likely identified residues that constitute the active site of FtsW. However, further characterization was necessary to rule out other possibilities.

**Table 1. Dominant-negative FtsW mutations isolated from the wrinkle-colony-based screen.**

| Mutation | # of isolates | Dominant-negative effect | Complementation | Location of the mutation |
|---|---|---|---|---|
| K133I | 1 | ++ | - | ECL2 |
| A135T | 2 | ++++ | - | ECL2 |
| A135V | 1 | ++++ | - | ECL2 |
| R137H | 1 | ++ | - | ECL2 |
| Q147E | 1 | ++++ | - | ECL2 |
| Q147K | 1 | ++ | - | ECL2 |
| P196T | 1 | ++ | - | ECL3 |
| L198I | 1 | + | - | ECL3 |
| G199A | 2 | +++ | - | ECL3 |
| Y242H | 1 | ++++ | - | ECL4 |
| R243H | 1 | +++ | - | ECL4 |
| R243L | 1 | ++++ | - | ECL4 |
| R243P | 1 | ++ | - | ECL4 |
| R243S | 1 | +++ | - | ECL4 |
| R246H | 2 | ++ | - | ECL4 |
| G371S | 1 | +++ | - | ECL5 |
| G381V | 2 | ++ | - | ECL5 |
| G381D | 1 | ++ | - | ECL5 |

Dominant-negative effect: "++++", "+++", "++" and "+" indicate that it takes ≤60 μM, 60–125 μM, ≥125 μM or ≥1000 μM IPTG to induce the expression of the mutant to completely block colony formation, respectively. Complementation: "-" indicates that the mutant cannot complement an *ftsW* depletion strain.

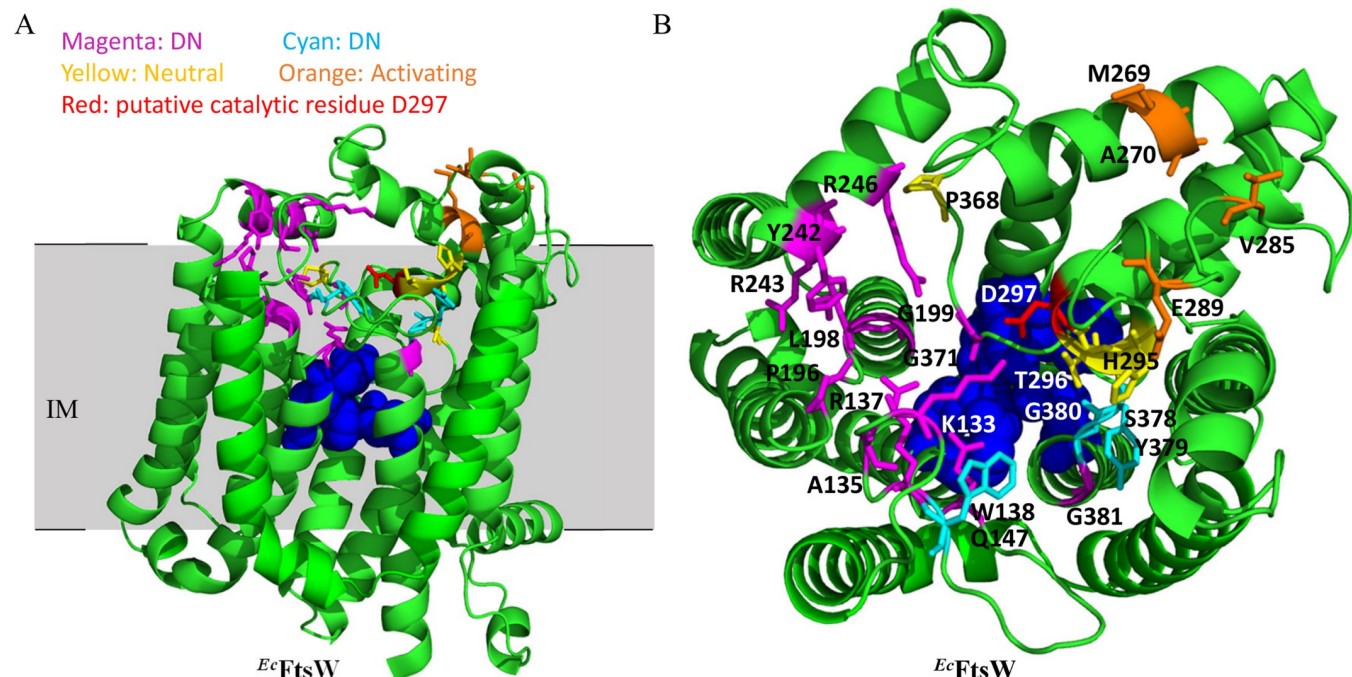

**Fig 3. Location of the dominant-negative *ftsW* mutations on a structural model of FtsW.** (A) and (B) Locations of dominant-negative mutations in a model of *E. coli* FtsW predicted by AlphaFold2 (45). Residues corresponding to the dominant-negative mutations are colored magenta (obtained by wrinkled-colony-based screen) or cyan (obtained by site-directed mutagenesis), whereas residues whose alanine substitution did not display a defect are colored yellow. The putative catalytic residue D297 is colored red, activating mutations are colored orange. Residues corresponding to those flanking the central cavity of RodA are colored blue and shown as spheres. The N-terminal 15 and C-terminal 10 amino acids are omitted.

**Table 2. Amino acid substitutions in FtsW generated by site-directed mutagenesis.**

| Mutation | Dominant-negative effect | Complementation | Location of the mutation |
|---|---|---|---|
| W138A | ++++ | - | ECL2 |
| E150A | - | + | ECL2 |
| H295A | - | + | ECL4 |
| T296A | - | + | ECL4 |
| D297A | ++ | - | ECL4 |
| P368A | - | + | ECL5 |
| S378A | - | + | ECL5 |
| Y379A | ++ | - | ECL5 |
| G380A | ++ | - | ECL5 |

Dominant-negative effect: "++++" and "++" indicate that it takes ≤60 μM or 60~125 μM IPTG to induce the expression of the mutant to completely block colony formation, respectively. "-" indicates the mutants do not display a dominant-negative effect. Complementation: "+" and "-" indicate that the mutant can or cannot complement an *ftsW* depletion strain, respectively.

## Isolation of additional dominant-negative FtsW variants in the ECLs by site-directed mutagenesis

The dominant-negative substitutions isolated in the wrinkled-colony-based screen are located in the ECLs around the central cavity, but the screen was not saturating. Therefore, we used site-directed mutagenesis to substitute additional highly conserved residues around the cavity in the ELCs of FtsW with alanine to see if they were dominant-negative (Table 2). As shown in Figs 4 and S4, three of the eight substitutions displayed a dominant-negative effect and no

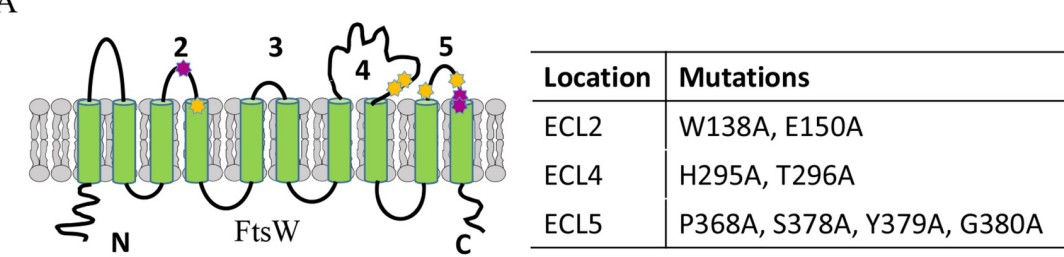

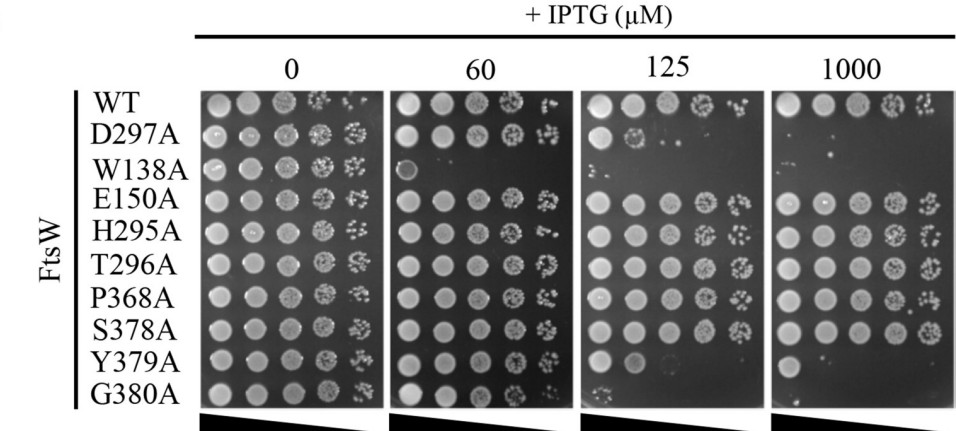

**Fig 4. Isolation of additional dominant-negative mutations of FtsW.** (A) Residues chosen for site-directed mutagenesis are indicated in the topology model of *E. coli* FtsW. Dominant-negative alleles are colored magenta, alleles that did not show an effect are colored yellow. (B) A spot test of the FtsW mutants. The test was performed as in Fig 1A.

longer complemented. Note that some of the mutants were more toxic or as toxic as the D297A mutant, indicating that dominant-negative mutations had a strong impact on FtsW's activity. However, the remaining substitutions were not dominant-negative and were still able to complement an *ftsW* depletion strain. These results suggest that not all of the highly conserved residues around the cavity are critical for FtsW function. Also, these results highlight the effectiveness of our wrinkled-colony-based screen to identify critical residues for the function of FtsW.

## Dominant-negative FtsW mutants block septal PG synthesis but not assembly of the divisome

Although the above analysis raises the possibility the dominant-negative mutations disrupt the active site of FtsW, there are other possibilities for their dominant-negative effect. One is that the variant was unable to localize to the Z ring, although it retained the ability to interact with a binding partner, such as FtsI, titrating it away from the septum. Although the location of the mutations suggested that this was unlikely, we checked the localization of the dominant-negative FtsW variants by fluorescence microscopy to rule out this possibility. As we obtained multiple dominant-negative mutations in 4 of the 5 ECLs of FtsW, we chose the 2 to 3 variants from each ECL that displayed the strongest dominant-negative effect for this analysis. FtsW variants were fused to GFP and expressed in an *ftsW* depletion strain at the non-permissive condition. As shown in Fig 5, cells harboring a vector expressing GFP became filamentous after FtsW was depleted by the temperature shift, whereas cells expressing FtsW-GFP divided normally with the protein localizing to the division site. Although cells expressing the dominant-negative FtsW variants (fused to GFP) became filamentous, the variants localized to potential division sites within the filaments, indicating they did not have a defect in localizing to the Z ring.

Another possibility was that the dominant-negative FtsW variants were unable to recruit downstream proteins to the Z ring to form a mature divisome complex, although they localized to the Z ring. To rule out this possibility, we checked the localization of FtsI and FtsN, which are recruited after FtsW and markers for a mature divisome, in cells overexpressing the dominant-negative FtsW variants. The test was carried out in a strain expressing ZapA-mcherry at its native locus [46], which interacts directly with FtsZ and has been widely used as a proxy for the Z ring. Microscopic inspection showed that overexpression of the dominant-negative FtsW variants in liquid cultures led to a division block and thus cell filamentation. Many filaments lacked ZapA-mcherry, GFP-FtsI or GFP-FtsN rings, presumably because the cells were dead. However, in the filaments with ZapA-mcherry rings, both GFP-FtsI and GFP-FtsN rings were frequently present at these potential division sites (Figs 6 and 7). Quantification of the co-localization of GFP-FtsI or GFP-FtsN with ZapA-mcherry showed that close to 70% of the ZapA-mcherry rings were associated with GFP-FtsI and GFP-FtsN rings before induction of the FtsW variants and the co-localization frequency was further increased after induction of the FtsW mutants (S5 Fig). This result indicated that dominant-negative FtsW variants did not interfere with the recruitment of these late components of the divisome.

Since the dominant-negative FtsW mutants did not have a defect in localizing to the Z ring and were able to recruit downstream proteins, we next tested if they were able to synthesize septal PG. To do this, we checked the incorporation of the fluorescent D-amino acid (FDAA) HADA at the division sites (marked by ZapA-mcherry) in cells overexpressing the dominant-negative FtsW mutants. FDAAs have been widely used to label sites of PG synthesis and their incorporation at the division site depends on active septal PG synthesis mediated by FtsW-FtsI [47–49]. As shown in Fig 8, a band of HADA was present at the constriction sites in cells

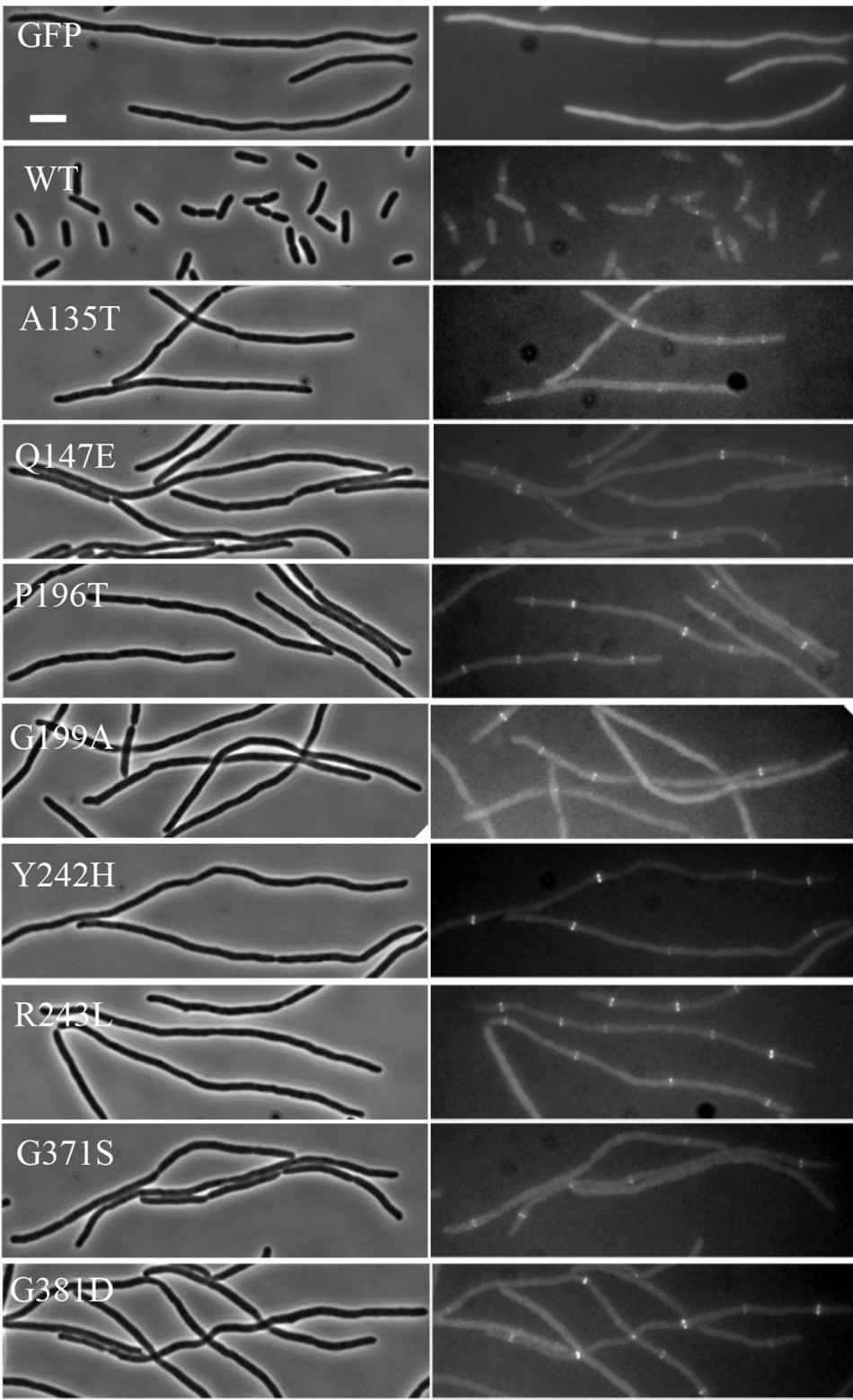

**Fig 5. Dominant-negative FtsW mutants localize to the Z ring.** Localization of the FtsW mutants was assessed in the FtsW depletion strain SD237 [W3110, *leu*::Tn*10 ftsW*::*kan*/pDSW406 (P<sub>BAD</sub>::*ftsW*)] using plasmid pSD349 (P$_{206}$::*ftsW-l60-gfp*) and derivatives carrying *ftsW* mutations. Overnight cultures of SD237 carrying derivatives of plasmid pSD349 were diluted 1:100 in fresh LB medium with antibiotics and arabinose. 2 hrs later arabinose was removed by

centrifugation and washing, and the cells were resuspended in fresh LB and diluted 1:20. IPTG was added to the culture to induce the fusion protein and 4 hours post induction (also removal of arabinose) cells were immobilized on a 2% agarose pad for photographing. Scale bar, 3 μm.

overexpressing wild type FtsW. However, HADA incorporation was absent at most of the potential division sites marked by ZapA-mcherry in filamentous cells overexpressing the dominant-negative FtsW mutants. Quantification of the results showed that co-localization of HADA bands and ZapA-mcherry rings dropped from 60–70% before induction of the FtsW mutants to less than 5% after induction of the FtsW mutants (S6 Fig), indicating that septal PG synthesis was blocked by the FtsW mutants. Collectively, these results show that the dominant-negative FtsW mutants do not have a defect in divisome assembly, but are unable to carry out septal PG synthesis.

## Dominant-negative FtsW mutants bind FtsI and lipid II *in vitro*

For FtsW to synthesize septal PG, it must bind its substrate lipid II and polymerize the disaccharide pentapeptide into glycan strands. In addition, previous studies showed that FtsW must bind FtsI as it is activated by a pathway that operates through its cognate bPBP (PBP3/FtsI) [30–32]. Thus, there are at least three possibilities for the observed defect in septal PG synthesis: inability to bind lipid II, a defect in catalysis or a defect in the activation step. Several lines of evidence suggest that these dominant-negative mutants are not defective in the activation step. First, as shown in Figs 1A and 2B an FtsW mutant defective in the activation step (M269K) was much less toxic than the putative catalytic mutant D297A. If the dominant-negative FtsW mutants were defective in activation, their toxicity was expected to be comparable to that of the M269K mutant. Instead, they were much more toxic than the M269K mutant and were comparable or more toxic than the presumptive catalytic D297A mutant, indicating that they were similar to D297A or belong to a different class. Second, it has been shown that residues critical for FtsW activation largely lie at the interaction interface between the ECL4 of FtsW and the pedestal domain of FtsI [32,40]. However, the isolated dominant-negative mutations were distributed in all the ECLs except for ECL1. Accordingly, in the predicted FtsW structure the dominant-negative mutations and the activating mutations reside at distinct locations (Fig 3). Third, while the activation defective M269K mutation could be suppressed by an activating mutation (E289G) of FtsW [32], none of the dominant-negative mutations could be suppressed by E289G as the double mutants were as toxic as the single mutants and unable to complement (S7 and S8 Figs). Therefore, although we could not exclude the possibility that some of the dominant-negative mutations may block the activation step, based on the reasoning above, we prefer the idea that they are either defective in substrate binding or catalysis.

To test substrate binding we co-purified 9 of the dominant-negative FtsW variants along with FtsI. To do this, His-FtsW was co-expressed with FtsI(PBP3) and membranes solubilized by dodecyl-β-D-maltopyranoside (DDM) and run on a HisTrap column to purify His-FtsW. As shown in Fig 9A, FtsI(PBP3) co-purified with all FtsW variants similarly to wild type FtsW. Incubation of the protein complexes with Bocillin showed that FtsI(PBP3) within the complexes bound Bocillin comparable to FtsI(PBP3) in the wild type complex. These results indicate that these FtsW variants bind to FtsI(PBP3) and do not significantly affect the ability of FtsI(PBP3) to bind its substrate. These results are also consistent with the observation that in cells overexpressing these FtsW variants FtsI(PBP3) was recruited to the Z ring (Fig 6). To ensure that the co-purification of the FtsW-FtsI(PBP3) complex was due to complex formation between FtsW and FtsI(PBP3) but not non-specific binding to the HisTrap column, we co-

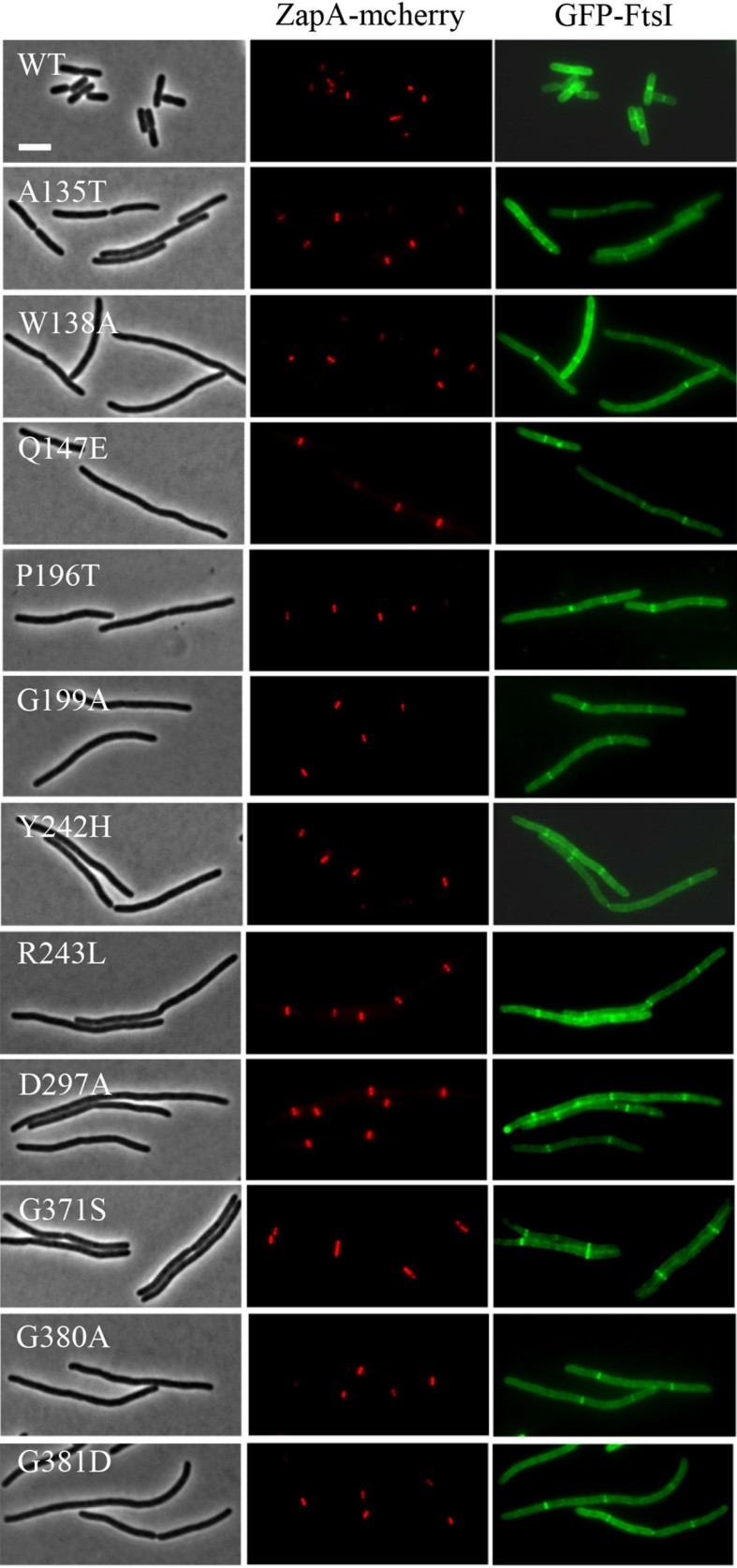

**Fig 6. Overexpression of the dominant-negative FtsW mutants does not prevent localization of GFP-FtsI.**
Overnight cultures of LYA4 (TB28, *zapA-mcherry cat<>frt*) carrying plasmid pLY114 (pBAD33, P_BAD:*gfp-linker-ftsI*)
were diluted 1:100 in fresh LB medium with antibiotics and cultured at 30˚C. 2 hours later the cultures were diluted
1:20, IPTG was added to a final centration of 250 μM to induce the expression of the FtsW mutants and arabinose was
added to a final concentration of 0.05% to induce expression of GFP-FtsI. 3 hours post induction cells were
immobilized on 2% agarose pads for photographing. Scale bar, 3 μm.

expressed HisFtsI(PBP3) with FtsW or FtsW_HA (a mutant form containing an HA tag inserted
between residues E293 and A294 of FtsW which prevents complex formation with FtsI [50]).
As shown in S9A and S9B Fig, FtsW but not FtsW_HA co-eluted with HisFtsI(PBP3), indicating
that complex formation was required for co-purification. As an additional control, FtsI(PBP3)
was co-expressed with another membrane protein (HisFtsN) and purified under the same con-
ditions as HisFtsW-FtsI(PBP3). In this case, HisFtsN was present in the elution fraction but
FtsI(PBP3) was present in the flow through (S9C Fig). Thus, co-purification of FtsI(PBP3)
with FtsW and the FtsW variants is mediated by its interaction with FtsW.

We next tested if the FtsW variants (in a complex with FtsI) bound their substrate lipid II
by a fluorescence anisotropy (FA) assay [51]. In this assay, nitrobenzoxadiazole (NBD)-labelled
lipid II was mixed with increasing concentrations of each of the HisFtsW-FtsI(PBP3) com-
plexes. Successful binding of NBD-lipid II by FtsW within the protein complex generates an
FA signal that increases until binding reaches saturation. This assay has been successfully used
to probe the binding of FtsW, PBP1b and MurJ to lipid II previously [51]. As shown in Fig 10,
all FtsW-FtsI(PBP3) variants displayed binding curves for NBD-lipid II similar to wild type
FtsW-FtsI(PBP3). Calculation of the disassociation constant for each of the FtsW-FtsI/PBP3
variants showed that the mutants had a similar or slightly higher affinity for NBD-lipid II.
These results suggest that none of the dominant negative FtsW variants had an obvious defect
in binding its substrate lipid II, indicating they were likely defective in catalysis. Despite exten-
sive efforts, however, we were unable to detect any enzymatic activity of the WT *E. coli*
FtsW-FtsI(PBP3) complex, making it impossible for us to confirm that these mutants were
defective for catalysis *in vitro*.

## Discussion

SEDS proteins are critical for PG synthesis during bacterial cell elongation and cell division.
Understanding how they interact with their binding partners, recognize their substrate lipid II
and polymerize it into glycan strands is necessary to elucidate the mechanisms regulating bac-
terial morphogenesis. In this study, we characterized a set of dominant-negative FtsW mutants
isolated using a wrinkled-colony-based screen. Extensive analyses of these mutants suggest the
corresponding residues identify the active site of FtsW which is very similar to that suggested
for RodA in previous studies. This finding will facilitate the study of FtsW and other SEDS
proteins as well as provide clues for the design of new antibiotics targeting this important class
of PG polymerases.

The isolation of dominant-negative mutants has recently been employed to probe the func-
tion of components of the elongasome and divisome [11,30,31,52]. For example, isolation and
characterization of dominant-negative MreC mutants and a subsequent screen for their sup-
pressors led to the discovery of the activation pathway governing elongasome activity [11,52].
Similarly, characterization of dominant-negative FtsL mutants revealed its critical role in acti-
vating FtsW-FtsI for septal PG synthesis [30,31]. In this study, we investigated the function of
FtsW by screening for dominant-negative mutants that was facilitated because they impaired
cell division and produced a winkled colony phenotype. This approach turned out be highly
effective and yielded a battery of dominant-negative FtsW mutants, the characterization of

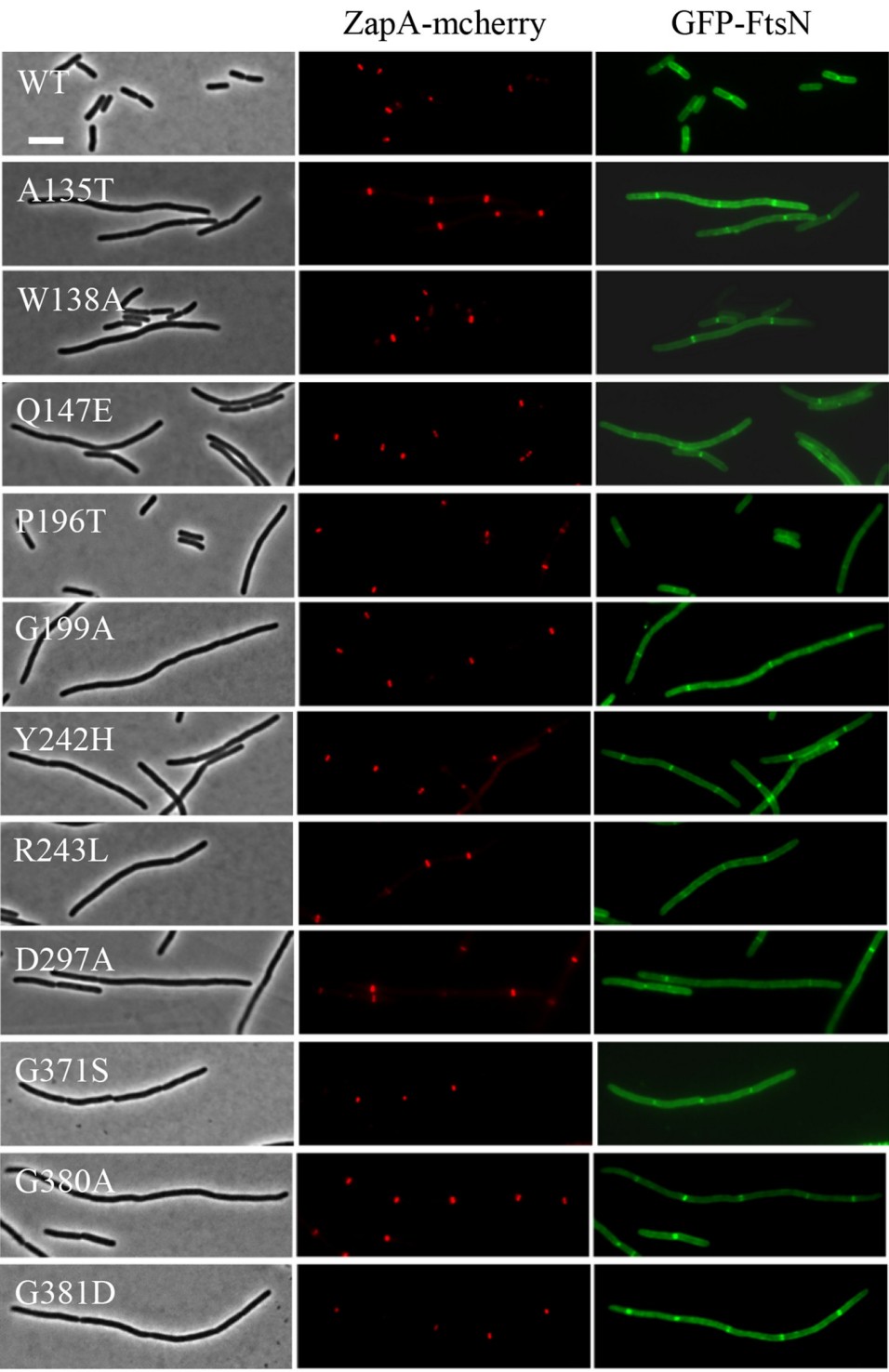

**Fig 7. Overexpression of the dominant-negative FtsW mutants does not prevent localization of GFP-FtsN.** The test was performed as in Fig 6 except plasmid pLY103 (pBAD33, P_{BAD}::*gfp-ftsN*) was used instead of pLY114. Scale bar, 3 μm.

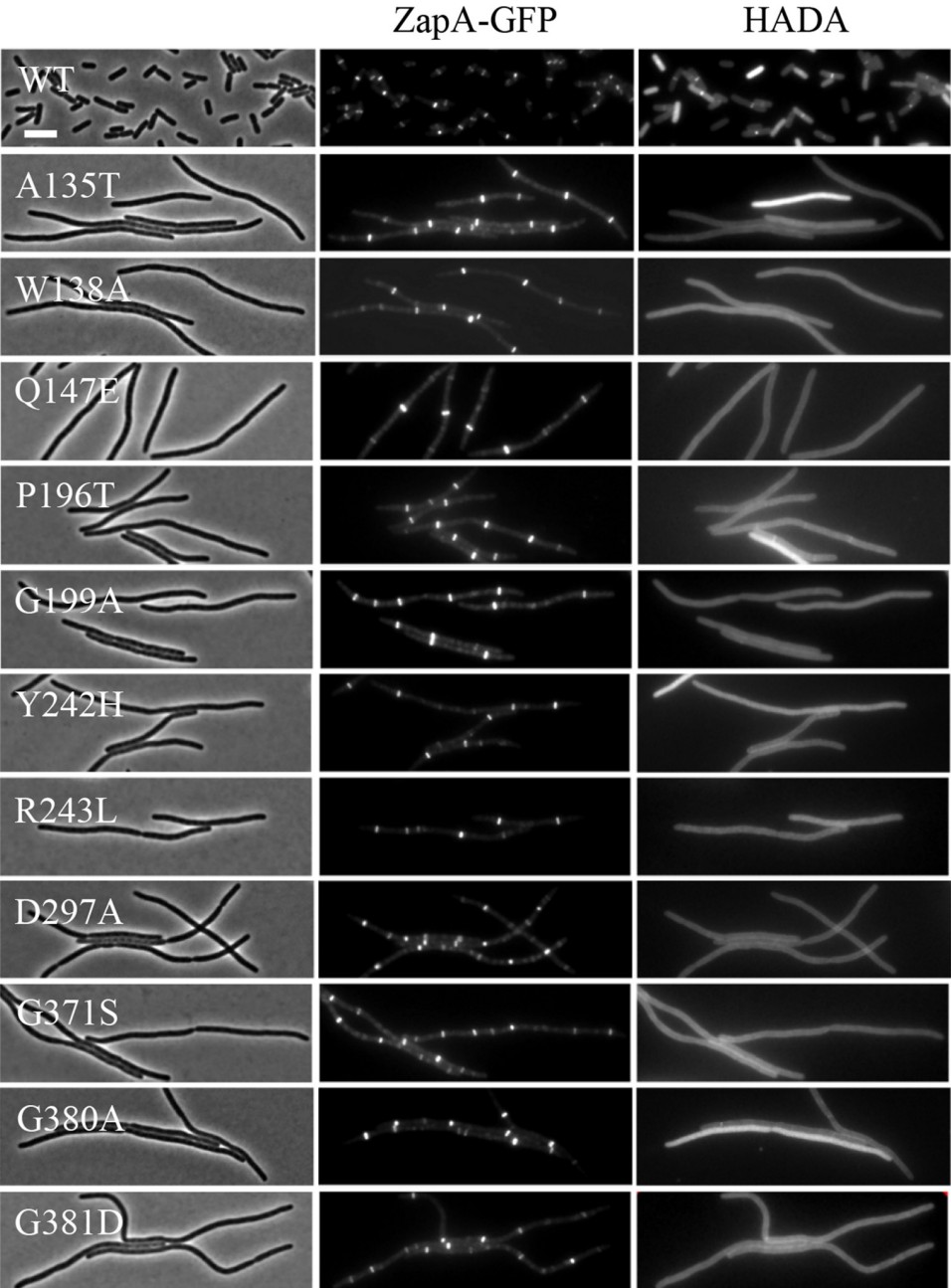

**Fig 8. Overexpression of the dominant-negative FtsW mutants blocks septal PG synthesis.** Overnight cultures of HC261 (TB28, *zapA-gfp*) carrying plasmid pSEB429 or its derivatives were diluted 1:100 in fresh LB medium with antibiotics and cultured at 30°C. 2 hours later the cultures were diluted 1:10 and IPTG was added to a final concentration of 200 μM to induce the expression of the FtsW mutants. 2 hours post induction, a 200 μl sample was taken from each culture and incubated with 2 μl of HADA (final concentration, 0.25 mM) in dimethyl sulfoxide (DMSO) for 1 min followed by fixation with paraformaldehyde and glutaraldehyde for 15 min on ice. Cells were then washed five times with phosphate-buffered saline (PBS) and resuspended in 50 μl of PBS and spotted onto an agarose pad for imaging. Scale bar, 3 μm.

which led to the identification of the presumptive active site of FtsW (discussion below). Many of the mutations correspond to critical residues in RodA, which were determined by mutagenesis followed by high-throughput sequencing (Mutseq). This suggests that our approach is

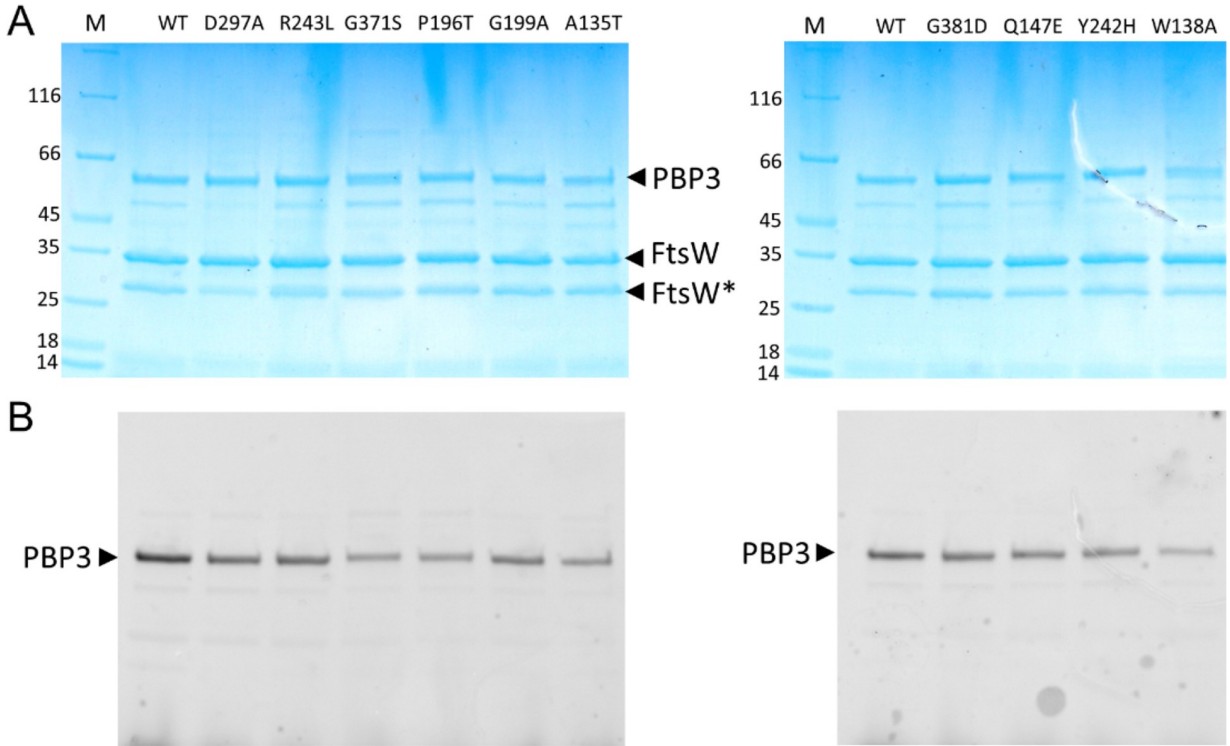

**Fig 9. Purification of FtsW variants in a complex with FtsI/PBP3.** FtsW and FtsI/PBP3 were co-expressed in *E. coli* cells; FtsW contains a His-Tag and was used as bait to co-purify untagged FtsI/PBP3 on a HisTrap column. The samples were incubated with Bocillin FL to label FtsI and analyzed by SDS-PAGE. The gels were first subjected to fluorescence imaging to detect Bocillin-bound FtsI/PBP3 (B) followed by Coomassie blue staining (A). The positions of FtsW and FtsI/PBP3 are indicated by arrows and FtsW mutations are shown on top of each lane of the gel. FtsW* indicates an FtsW degradation product. M, protein standards.

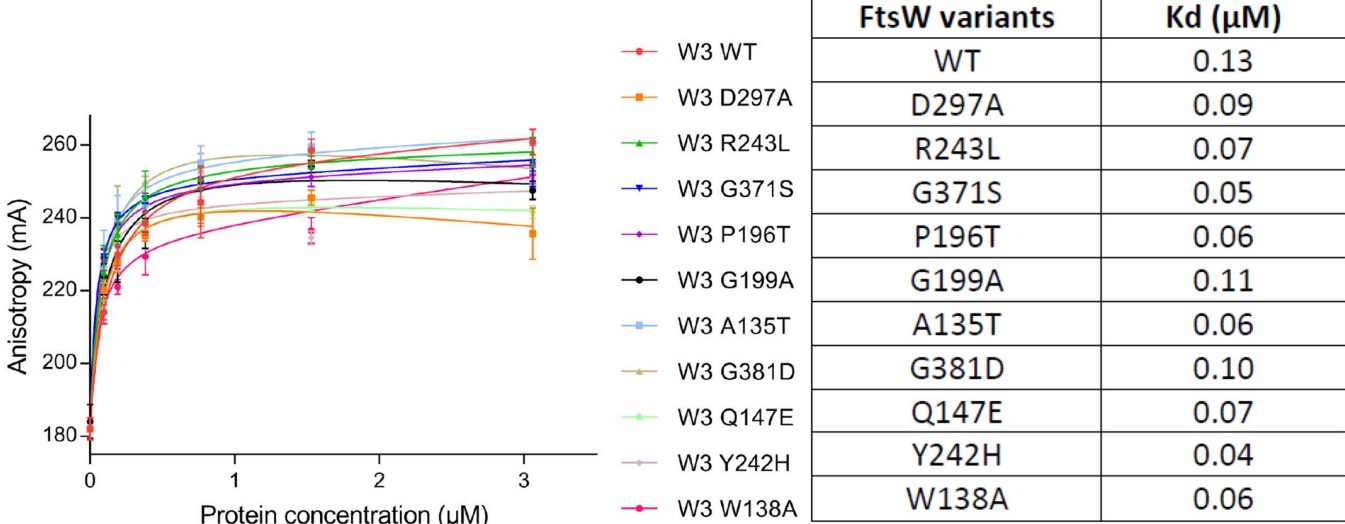

**Fig 10. Direct binding of the FtsW variants to NBD-lipid II assayed by a fluorescence anisotropy (FA) assay.** The assay was performed as described in Materials and Methods. FA (in mA units) is plotted as a function of FtsW-FtsI/PBP3 complex concentrations. FtsW-FtsI/PBP3 is indicated by "W 3". The error bars represent the FA values as mean ± s.d. of triplicate experiments. The disassociation constant Kd (μM) of FtsW-FtsI/PBP3 variants for NBD-lipid II was determined with Graphpad Prism 6.0 and is indicated to the right.

comparable to Mutseq in identifying essential residues of proteins of interest. We propose that the wrinkled-colony-based screen is a highly effective approach for identification of critical residues of cell division proteins that form a complex and should be applicable to other proteins and many other bacterial species.

Because mutations in the RodA-PBP2 interaction interface in the membrane plane has been shown to be dominant-negative and structures of RodA alone or in complex with PBP2 revealed two possible substrate binding sites [9,10], we expected to isolate dominant-negative alleles of FtsW that affect a variety of its functions, such as interaction with FtsI, activation in response to FtsN, and substrate binding. However, characterization of the alleles we isolated suggest that they all disrupt the catalytic activity of FtsW. It is surprising that we did not get any mutations affecting these other functions. We also did not obtain previously reported dominant-negative mutants R146A and K153A which were originally thought to affect the flippase activity of FtsW [50]. One possibility is that our screen for the dominant-negative mutants was not saturating so that these mutants did not show up. Another possibility is that FtsW mutants that do not bind lipid II or FtsI are not recruited as well to the Z ring and therefore are unable to displace the WT protein. A third possibility, which is not mutually exclusive, is that carrying out the wrinkled colony selection at such a low expression level of the mutant protein yields only very toxic mutants. Perhaps screening for wrinkled-colonies at higher inducer concentrations would yield additional classes of mutants that affect other activities of FtsW.

Several lines of evidence suggest that the residues corresponding to the dominant-negative mutations constitute the active site of FtsW (Table 3). First, most of the mutations affect highly conserved residues that reside in the ELCs of FtsW. Moreover, mapping of these mutations to an FtsW structure produced by AlphaFold2 showed that they cluster around the central cavity facing the periplasm, which has been suggested as the active site of SEDS proteins [9]. Microscopic analysis showed that the dominant-negative mutations did not affect the localization of FtsW to the Z ring or its ability to recruit downstream proteins FtsI and FtsN. In addition, the mutations did not affect the interaction between FtsW and its substrate lipid II nor could they be rescued by a strong activation mutant that bypasses the activation pathway elicited by FtsN. Thus, the dominant-negative FtsW variants appear specifically defective in catalysis. Consistent with this interpretation, overexpression of these dominant-negative mutants blocked septal PG synthesis mediated by FtsW-FtsI *in vivo*. Given that the positions of the mutations are

**Table 3. Phenotypes of dominant-negative FtsW variants.**

| Mutation | Localization to midcell | Recruit FtsI & FtsN | Synthesize septal PG | Substrate binding | FtsI binding | Synthesize PG *in vitro* |
|:---:|:---:|:---:|:---:|:---:|:---:|:---:|
| WT | + | + | + | + | + | - |
| A135T | + | + | - | + | + | ND |
| W138A | + | + | - | + | + | ND |
| Q147E | + | + | - | + | + | ND |
| P196T | + | + | - | + | + | ND |
| G199A | + | + | - | + | + | ND |
| Y242H | + | + | - | + | + | ND |
| R243L | + | + | - | + | + | ND |
| D297A | + | + | - | + | + | ND |
| G371S | + | + | - | + | + | ND |
| G381D | + | + | - | + | + | ND |

"+" or "-" indicates that the mutants are positive or negative in the indicated tests. "ND" means not determined.

close to the interaction interface between FtsW and FtsI, and the ECLs of FtsW are highly dynamic, we cannot exclude the possibility that some of the mutations may also affect the interaction with FtsI or the allosteric activation of FtsW. Nonetheless, based on our *in vivo* and *in vitro* characterization of the mutants, it is tempting to speculate that residues altered by the dominant-negative mutations constitute the active site of FtsW. Unfortunately, we were unable to confirm that the FtsW variants were inactive for PG polymerization *in vitro* since a purified FtsW-FtsI complex from *E. coli* did not display any activity *in vitro*. Also, it is note-worthy to point out that similar to other family of PG polymerases, SEDS proteins should have two binding sites for lipid II, a donor site and an acceptor site [53]. A single mutation affecting one of the two binding sites may not display a strong binding defect *in vitro*. Thus, although we favor the idea that the dominant-negative mutations specifically affect catalysis by FtsW, it is possible some of the mutations may affect substrate binding which was not revealed by our FA assay. Despite this uncertainty, identification of these residues critical for FtsW activity *in vivo* should be useful for further biochemical and structural analysis of FtsW and other SEDS proteins as well as for designing inhibitors of SEDS proteins.

In summary, we utilized a wrinkled-colony-based screen to isolate dominant-negative vari-ants of the septal PG polymerase FtsW that likely indicate its active site. We expect that the wrinkled-colony-based screen will facilitate the study of cell division proteins in many species and the determination of the potential active site of FtsW will aid the study of SEDS proteins and development of their inhibitors.

## Materials and methods

### Media, bacterial strains, plasmids and growth conditions

Cells were grown in LB medium (1% tryptone, 0.5% yeast extract, 0.5% NaCl and 0.05 g/L thy-mine) at indicated temperatures. When needed, antibiotics were used at the following concen-trations: ampicillin = 100 μg/ml; spectinomycin = 25 μg/ml; kanamycin = 25 μg/ml; tetracycline = 12.5 μg/ml; and chloramphenicol = 15 μg/ml. Strains, plasmids and primers used in this study are listed in S1, S2 and S3 Tables, respectively. Construction of strains and plasmids is described in detail in S1 Text with the primers listed in S3 Table. The fluorescent D-amino acid HADA was purchased from the company Scilight-Peptide (http://www.scilight-peptide.com/).

### Creation of an FtsW mutant library and screen for dominant-negative FtsW mutants

Construction of the FtsW mutant library has been described previously [32]. Briefly, an error-prone PCR-mutagenized copy of *ftsW* was cut and ligated into the plasmid pSEB429 cut with the same enzymes (pDSW208, $P_{204}$::*ftsW*) to replace the wild type *ftsW*. Ligation products were transformed into JS238 competent cells and plated on LB plates with ampicillin and glu-cose. About 20,000 transformants were pooled together and plasmids were isolated and stocked.

To screen for dominant-negative FtsW mutants, transformants were selected on LB plates with ampicillin and 30 μM IPTG. Expression of dominant-negative FtsW mutants at this con-centration of IPTG partially inhibits cell division, resulting in cell chaining and filamentation which causes the appearance of flat, wrinkled colonies with rough edges. These wrinkled colo-nies appeared at a 3–5% frequency and could be easily identified by eye. Wrinkled colonies were randomly picked and restreaked on LB plates with or without 30 μM IPTG to confirm that the wrinkled-colony phenotype was IPTG-dependent. Cells from the plate with 30 μM

IPTG were examined under a microscope to confirm that cell division was partially inhibited. If a colony passed this test, the plasmid was isolated and transformed back to JS238 to confirm the dominant-negative effect was linked with the plasmid. The *ftsW* gene in the plasmid was then sequenced to identify the mutation(s). In cases where the plasmid harbored multiple mutations, each mutation was introduced into the parental plasmid pSEB429 using site-directed mutagenesis and tested for a dominant-negative effect to identify the causative mutation. 25 wrinkled colonies were randomly picked and analyzed, leading to the identification of 20 dominant-negative FtsW alleles.

## Fluorescence microscopy

All phase contrast and fluorescence images were acquired using an Olympus BX53 upright microscopes with a Retiga R1 camera from QImaging, a CoolLED pE-4000 light source and a U Plan XApochromat phase contrast objective lens (100X, 1.45 numerical aperture [NA], oil immersion). Green, mcherry and HADA (blue) fluorescence was imaged using the Chroma EGFP filter set EGFP/49002, mcherry/Texas Red filter set mcherry/49008 and the DAPI filter set DAPI/49000, respectively. The procedure for each individual experiment was described as followed.

**Localization of FtsW-L60-GFP and its mutants.** Overnight cultures of SD237 [W3110, *leu*::*Tn10 ftsW*::*kan*/pDSW406 (P$_{BAD}$::*ftsW*)] carrying plasmid pDSW210 (P$_{206}$::*gfp*) or pSD247 (P$_{206}$::*ftsW-l60-gfp*) and its derivatives were diluted 1:100 in fresh LB medium with antibiotics and 0.2% arabinose, and grown at 30˚C for 2 h. Cells were then collected by centrifugation and washed twice with fresh LB to remove the arabinose, followed by resuspension in the same volume of LB medium. These arabinose-free cultures were then diluted 1:20 in fresh LB medium and IPTG was added to a final concentration of 60 μM. 4 hours post removal of arabinose and induction with IPTG, cells were immobilized on 2% agarose pads for imaging with an exposure time of 1 second.

**Localization of GFP-FtsI and GFP-FtsN.** Overnight cultures of LYA4 (TB28, *zapA-mcherry cat<>frt*) carrying plasmid pLY114 (pBAD33, P$_{BAD}$::*gfp-ftsI*) or pLY103 (pBAD33, P$_{BAD}$::*gfp-ftsN*) and pSEB429 (pDSW208, P$_{204}$::*ftsW*) or its derivatives from 30˚C were diluted 1:100 in fresh LB medium with antibiotics, grown at 30˚C for 2 h. The cultures were then diluted 1:10 in fresh LB medium with antibiotics, 0.05% arabinose and 200 μM IPTG and grown at 30˚C for 2.5 hours. Cells were immobilized on 2% agarose pad for imaging with an exposure time of 1 second.

**HADA labeling to determine septal PG synthesis.** To check if expression of the dominant-negative FtsW mutants blocked HADA incorporation (septal PG synthesis), overnight cultures of HC261 cells harboring plasmid pSEB429 (pDSW208, P$_{204}$::*ftsW*) or its derivatives were diluted 1:100 in fresh LB medium with antibiotics, and grown at 30˚C for 2 h. The cultures were then diluted 1:10 in fresh LB medium with antibiotics and split into two parts. IPTG was added to a final concentration of 200 μM to one part and then grown at 30˚C for another 2 hours. A 200-μl sample was then taken from each culture and incubated with 2 μl of HADA (final concentration, 0.25 mM) in dimethyl sulfoxide (DMSO) for 1 min. After the incubation, the cells were immediately fixed with paraformaldehyde and glutaraldehyde for 15 min on ice and were then washed five times with phosphate-buffered saline (PBS). The cells were then resuspended in 50 μl of PBS and were spotted onto an agarose pad for imaging with an exposure time of 1 second.

## Purification of FtsW-PBP3 complexes

*E. coli* FtsW-FtsI/PBP3 wild-type complex and mutants were expressed in *E. coli* strain C43 (DE3) harboring plasmid pDML2041 (containing HisFtsW and FtsI) as previously described

[54]. Briefly; bacteria were grown at 37˚C, in LB medium supplemented with 100 μg/ml ampicillin to an $A_{600nm}$ of 0.6. Expression was induced for 4–5 hours by addition of 1 mM isopropyl β-D-1-thiogalactopyranoside (IPTG). The cell membranes were isolated and solubilized in 25 mM Tris-HCl, pH 8.0, 500 mM NaCl, 10% glycerol 40 mM *n*-dodecyl-β-D-maltopyranoside (DDM, Inalco) and an EDTA-free protease inhibitors cocktail (Roche) for 1 hour at room temperature. The mixture was centrifuged at 150,000 x g for 1 hour at 4˚C and the supernatant containing solubilized membrane proteins collected. The FtsW-FtsI/PBP3 complex was purified on a HisTrap column (GE HealthCare) conditioned in 50 mM Tris-HCl, pH 7.5, 300 mM NaCl, 10% glycerol, 1 mM DDM and 50 mM imidazole. The proteins were eluted with an imidazole gradient (0.05–1 M) and the fraction analyzed by SDS-PAGE. The pure fractions were pooled and desalted on a G25 Sephadex column using buffer A (50 mM Hepes, pH 7.5, 300 mM NaCl, 10% glycerol and 1 mM DDM. The proteins were concentrated using an Amicon apparatus (EMD Millipore) with a 100 kDa cutoff membrane and stored at −20˚C. The concentration of the proteins was determined with the aid of the BCA reagent kit (Thermo-Fisher Scientific).

## Fluorescent anisotropy assay

FA experiments were performed to test for the interaction of FtsW with a fluorescent lipid II as described previously [51]. Serial dilutions of wild-type and FtsW-mutants in complex with FtsI/PBP3 in the buffer (50 mM Hepes pH 7.5, 0.1M NaCl, 0.005% DDM, 0.2% CHAPS) were prepared in 384-well plates, and the probe was added at 0.33 μM final concentration in a final volume of 30 μl. The mixtures were incubated for 2–30 min at 21˚C, and the FA signals were recorded using an Infinite F Plex (Tecan, Männedorf, Switzerland) equipped with polarization filter with excitation wavelength at 485 nm and emission at 535 nm. FA values were calculated using the equations $FA = (I_\parallel - G{\cdot}I_\perp)/I_\parallel + 2G{\cdot}I_\perp)$, where $I_\parallel$ is the fluorescence intensity of emitted light parallel to excitation, $I_\perp$ is the fluorescence intensity of emitted light perpendicular to excitation, and $G$ is the correction factor that correct for instrument bias. The $G$ factor is experimentally determined using the probe alone. For $K_d$ determinations, the fluorescence anisotropy data were analyzed by nonlinear curve fitting using GraphPad Prism 6.0 software as described [51].

## Supporting information

**S1 Text. Construction of strains and plasmids.**
(DOCX)

**S1 Table. Bacterial strains used in this study.**
(DOCX)

**S2 Table. Plasmids used in this study.**
(DOCX)

**S3 Table. Primers used in this study.**
(DOCX)

**S1 Fig. Complementation test of the dominant-negative FtsW variants isolated by the wrinkled-colony-based screen.** Plasmids pDSW208, pSEB429 (pDSW08, P₂₀₄::*ftsW*) or its derivatives harboring a dominant-negative *ftsW* mutation were transformed into strain SD237 [W3110, *leu*::Tn10 *ftsW*::*kan*/pDSW406 (pBAD33, P_BAD::*ftsW*)] on LB plates with ampicillin and 0.2% arabinose. The next day, a single transformant of each resulting strain was resuspended in 1 ml of LB medium, and serially diluted ten-fold. 3 μl of each dilution was spot on

LB plates with antibiotics, with or without increasing concentrations of IPTG. Plates were incubated at 30˚C overnight and imaged.
(TIF)

**S2 Fig. Alignment of the ELCs of FtsW and RodA from diverse bacterial species.** Amino acid sequences of FtsW and RodA were obtained from NCBI, aligned with Clustal Omega and then depicted using ESPRIPT: http://esprit.ibcp.fr/. Residues were numbered according to the *E. coli* FtsW sequence and those corresponding to the dominant-negative mutations are indicated by blue triangles. FtsW: *E. coli* (gi|2132970), *K. pneumoniae* (gi|597728208), S. flexneri (gi|110613701), *Y. pestis* (gi|115346361), *B. thailandensis* (gi|685745844), S. enterica (gi|205337406), *P. aeruginosa* (gi|15599609), *L. pneumophila* (gi|295650162), *M. xanthus* (gi|108761950), *C. crescentus* (gi|426019958), *A. tumefaciens* (gi|586950133), *B. fragilis* (gi|598888368), *B. subtilis* (gi|2493592), *L. monocytogenes* (gi|424013134), *E. faecalis* (gi|323480530), *S. aureus* (gi|384230086), *M. tuberculosis* (gi|613782430), *C. glutamicum* (gi|674168391), *B. burgdoferi* (gi|2493585), *T. thermophiles* (BAW00664.1). RodA: *E. coli* (gi|78101784), *S. flexneri* (gi|78101785), *K. pneumonia* (gi|499531772), *S. enterica* (gi|16501890), *P. aeruginosa* (gi|15599197), *L. pneumophila* (gi|148280678), B. thailandensis (gi|83652485), *V. cholera* (gi|126519168), *Y. pestis* (gi|115348295), *C. crescentus* (gi|220963725), *B. subtilis* (gi|732351), *M. xanthus* (gi|108462975), *L. monocytogenes* (gi|336024336), *S. pneumoniae* (gi|302638595), *M. tuberculosis* (gi|444893486), *C. glutamicum* (gi|62388939), *T. thermophilus* (WP_011228544.1).
(TIF)

**S3 Fig. Residues critical for *B. subtilis* RodA function and their location in a model of the *B. subtilis* RodA structure predicted by AlphaFold2.** Mutability of critical residues of *B. subtilis* RodA were categorized based on previous Mutseq analysis [5]. Dominant-negative *E. coli* FtsW mutations isolated in this study that alter identical residues in *B. subtilis* RodA are indicated in parentheses. Residues critical for *B. subtilis* function are mapped to a model of RodA and the residues of FtsW whose substitutions displayed a dominant-negative effect are mapped to a model of FtsW. Red: immutable residues; magenta: residues that tolerate only conservative changes (dominant-negative mutations in FtsW); pink: residues with limited mutability.
(TIF)

**S4 Fig. Complementation test of the FtsW variants generated by site-directed mutagenesis.** Transformation and spot tests were performed as in S1 Fig.
(TIF)

**S5 Fig. Quantification of co-localization of ZapA and FtsI or FtsN in cells overexpressing FtsW variants.** ZapA-mcherry rings and GFP-FtsI or GFP-FtsN rings displayed in Fig 6 and 7 were identified manually using ImageJ software. 50–200 ZapA-mcherry rings were examined for each strain and condition and the associated GFP-FtsI rings or GFP-FtsN rings were counted and plotted.
(TIF)

**S6 Fig. Quantification of co-localization of ZapA and HADA labeling in cells overexpressing FtsW variants.** ZapA-mcherry rings and HADA bands represented in Fig 8 were identified manually using ImageJ software. 50–200 ZapA-mcherry rings were examined for each strain and condition and the associated HADA bands were counted and plotted.
(TIF)

**S7 Fig. Toxicity test of the dominant-negative *ftsW* mutations in combination with the activating mutation E289G.** Transformation and spot test was performed as in Fig 1A. None

of the dominant-negative *ftsW* mutations was suppressed by E289G.
(TIF)

**S8 Fig. Complementation test of the ability of the activating mutation E289G to suppress the dominant-negative *ftsW* mutations.** Transformation and spot test was performed as in S1 Fig. None of the dominant-negative *ftsW* mutations was suppressed by E289G.
(TIF)

**S9 Fig. Control experiments for the purification of HisFtsW-FtsI/PBP3 shown in Fig 9.** Protein-protein interaction was assessed by co-expression and co-purification of His-tagged protein with untagged protein as indicated above each panel. The proteins are co-expressed in *E. coli* and the membrane fractions isolated and solubilized by DDM detergent followed by purification on HisTrap column. The panels **A-C** show the SDS-PAGE analysis of the elution fractions. (**A**) Untagged FtsW co-elutes with His-tagged FtsI/PBP3. (**B**) Untagged FtsW$_{HA}$, which does not bind to FtsI/PBP3, is not retained on a HisTrap column and was detected in the flow through using anti-HA antibodies. (**C**) When untagged FtsI/PBP3 was co-expressed with HisFtsN and purified in the same condition as HisFtsW-FtsI/PBP3, only HisFtsN was present in the elution fraction and PBP3 was detected in the unbound (Ub) fractions (detect by using Bocillin labelling (Fluo)), indicating no interaction between HisFtsN and PBP3. M, protein standard; E, elution fractions; Ft, flow through; FtsW$^*$ is a degradation product of FtsW. FtsW$_{HA}$ contains an insertion of a nine amino acid hemagglutinin (HA) peptide in the large loop between TM7 and TM 8. α-HA, immunoblot analysis using antibodies against the HA epitope of FtsW$_{HA}$. Ex, membrane extraction fraction. Fluo, fluorescence analysis of PBP3 labeled with Bocillin.
(TIF)

## Acknowledgments

We thank members of the Du lab, Terrak lab and Lutkenhaus lab for comments and advice in preparing the manuscript.

## Author Contributions

**Conceptualization:** Adrien Boes, Joe Lutkenhaus, Mohammed Terrak, Shishen Du.

**Data curation:** Ying Li, Mohammed Terrak, Shishen Du.

**Formal analysis:** Ying Li, Adrien Boes, Joe Lutkenhaus, Mohammed Terrak, Shishen Du.

**Funding acquisition:** Adrien Boes, Joe Lutkenhaus, Mohammed Terrak, Shishen Du.

**Investigation:** Ying Li, Adrien Boes, Yuanyuan Cui, Shan Zhao, Qingzhen Liao, Han Gong, Mohammed Terrak, Shishen Du.

**Methodology:** Ying Li, Adrien Boes, Yuanyuan Cui, Shan Zhao, Qingzhen Liao, Han Gong, Eefjan Breukink, Mohammed Terrak.

**Project administration:** Adrien Boes, Joe Lutkenhaus, Mohammed Terrak, Shishen Du.

**Resources:** Eefjan Breukink, Joe Lutkenhaus, Mohammed Terrak, Shishen Du.

**Supervision:** Adrien Boes, Joe Lutkenhaus, Mohammed Terrak, Shishen Du.

**Validation:** Adrien Boes, Han Gong.

**Visualization:** Ying Li, Adrien Boes.

**Writing – original draft:** Ying Li, Shishen Du.

**Writing – review & editing:** Ying Li, Adrien Boes, Eefjan Breukink, Joe Lutkenhaus, Mohammed Terrak, Shishen Du.

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
