## [Decision Letter · Decision Letter 0]

27 Oct 2021

Dear Dr Du,

Thank you very much for submitting your Research Article entitled 'Identification of the potential active site of the septal peptidoglycan polymerase FtsW' to PLOS Genetics.

The manuscript was fully evaluated at the editorial level and by independent peer reviewers. The reviewers appreciated the attention to an important problem, but raised some substantial concerns about the current manuscript. Based on the reviews, we will not be able to accept this version of the manuscript, but we would be willing to review a much-revised version. We cannot, of course, promise publication at that time.

If you decide to revise the manuscript for further consideration at PLOS Genetics, please aim to resubmit within the next 60 days, unless it will take extra time to address the concerns of the reviewers, in which case we would appreciate an expected resubmission date by email to plosgenetics@plos.org.

[LINK]

We are sorry that we cannot be more positive about your manuscript at this stage. Please do not hesitate to contact us if you have any concerns or questions.

Yours sincerely,

Daniel B. Kearns

Associate Editor

PLOS Genetics

Lotte Søgaard-Andersen

Section Editor: Prokaryotic Genetics

PLOS Genetics

The reviewers (and I) were enthusiastic about the potential impact of the work but both requested similar modifications prior to further consideration. In particular, the reviewers would like to see quantitation of the cell biological results, a request that seems reasonable and actionable. In addition, they felt that additional controls were required to interpret some of the biochemical analyses, particularly a negative control for the pull-down assay. I support both requests and think a mutant that disrupts the the FtsW/FtsI interaction would be ideal. Finally, there was some concern about the interpretation of the active site nature of the mutations. I realize that there are technical limitations to directly testing activity and that the interpretation is largely a discussion point anyway, but in revision, perhaps tread more carefully in the interpretation and/or present alternative possibilities. I would like to consider the manuscript further, if you feel the comments can be experimentally addressed.

Reviewer's Responses to Questions

**Comments to the Authors:**

Reviewer #1: In this manuscript by Li & Boes, the investigators present a genetic analysis of E.coli FtsW, a SEDS enzyme responsible for polymerization during division. FtsW works together with its enzymatic partner FtsI (also called PBP3) in the context of a multi-protein assembly, the divisome, to accomplish PG synthesis. While the polymerase activity of FtsW is well-established at this point, the molecular details of its enzymatic mechanism remain incompletely understood. Specifically, the mechanism by which the enzyme interacts with its lipid II substrate and accomplishes glycan chain polymerization is largely unknown. To identify residues involved in catalysis, the authors screen a library of FtsW mutants using a dominant negative assay based on a wrinkled colony phenotype. This strategy ensures that the identified mutations do not impair protein function at the level of expression or folding. Genetic analysis reveals a handful of mutants in the extracellular loops of FtsW as well as in the central “cavity” of the protein that inhibit division, consistent with previous reports on the homologous glycosyltransferase RodA. The authors suggest that nonfunctional mutants of FtsW preclude its catalytic activity rather than localization in the cell, since all of the identified mutants successfully localize to the division site but fail to produce PG in vivo. The authors then use a fluorescence anisotropy assay to probe lipid II binding to show that mutations do not affect lipid II binding, and so, the authors argue, must instead disrupt polymerization.

Overall, the isolation and validation of the mutants using in vivo assays is carefully performed and presented clearly. A major shortcoming of the manuscript is that the exact cause of the dominant negative effect of the mutations is not definitively established. Evidence is presented showing that localization, lipid binding, and downstream protein recruitment are intact in the mutants, leaving defects in catalysis as the most likely explanation. However, this is not directly assessed or demonstrated, and the authors state that they were unable to establish a polymerization assay for WT FtsW/FtsI complex. The result is that the data presented, while interesting, are incomplete for fully understanding what is happening with the mutants.

Main issues:

1. In figures 5-8, the authors present imaging data to show localization of various divisome components to the septum. These representative images are compelling and consistent with the scientific conclusions of the manuscript. However, it would be useful to quantify the colocalization of the Z-ring with the proteins of interest (or HADA signal) and verify that the effect holds on a large dataset. This is particularly important in Figure 8, since the HADA signal appears variable in both the WT and FtsW mutant cells.

2. Do the authors believe that the presented colocalization experiments can definitively prove that the divisome complex is fully assembled in the context of the FtsW mutants? If not, conclusions about the effect of mutations on divisome assembly should be toned down.

3. In figure 9, the authors use co-expression and a pulldown assay via Nickel-NTA resin to show that FtsW and PBP3 retain a stable complex, even in the context of nonfunctional mutations. However, by eye, the ratios of FtsW-to-PBP3 appear to be somewhat variable. A quantification of the populations of the complex vs dissociated components by size-exclusion chromatography would provide a more rigorous analysis of the binding interaction. Additionally, the pulldown experiment should include as a negative control a lane for the FtsI expressed in the absence of FtsW. Many proteins (particularly penicillin-binding proteins) can stick non-specifically to NTA resin. A related minor point is that the description of the pull-down in the main text is very brief, and providing slightly more detail here would be helpful.

4. The authors argue that their dominant-negative mutants are not defective in the “activation” step of the catalysis, and present three arguments to support their claims. However, counter arguments can be made for each case.

a. A well-characterized mutant (M269K) that is defective in activation is less toxic than the newly isolated mutants. However, the relative toxicity of the variants could be explained by differences in the degree of activation (or expression, degree of foldedness, etc.) rather than differences in the mechanism of disruption.

b. The position of the mutants (some of them are in the “cavity” rather than facing the extracellular interface of FtsW) is not consistent with disruption of FtsW-PBP3 interactions, as reported previously for RodA and its cognate enzymatic partner PBP2. However, the fact that known residues involved in allostery between these proteins tend to cluster in one location does not exclude the possibility that other regulatory residues are yet to be identified. Another important consideration is that a number of identified mutants do map to the extracytoplasmic loops of FtsW in the AlphaFold model of the protein. Given that these loops are likely to be dynamic, it is difficult to determine a priori what their orientation and function would be in the PBP3-bound state of FtsW - facing the inner cavity (presumably, catalytic), or at the interface with PBP3 (presumably, critical for binding of the two components), or perhaps both.

c. Suppression of M269K defects by the E289G mutation in FtsW may be specific to the interaction of these residues and may not apply globally to other residues involved in activation.

5. In figure 10, the authors measure lipid II binding using a fluorescence anisotropy binding assay. The intrinsic properties of the lipid II molecule as well as detergent solubilized proteins make them prone to non-specific interactions. Thus, it is difficult to verify the results of the lipid II binding assay in the absence of internal negative controls. If there are known mutations that disrupt lipid II binding in FtsW or lipid II analogs with similar chemical properties, the authors can use either a mutant protein or a non-interacting lipid analog as a negative control. In the absence of these tools, the authors should include a non-binding membrane protein control to validate their approach.

6. As mentioned above, the fact that purified WT FtsW-PBP3 has no detectable polymerization activity is highly problematic and calls into question the results of other in vitro assays (particularly, the lipid II binding assay). Additionally, since the main claim of the manuscript is that the identified mutants are deficient in polymerization, it is critical that the authors are able to probe catalytic activity directly.

Minor issues:

1. In Figures 5-8, the mutants used in the assays were not the same. Why were different mutants included in Figure 5 than in 6-8? Of particular note is the lack of the D297A mutant, which is a control used in most other assays.

2. In the main text, please include a description of ZapA-mCherry and its use as a proxy for the presence of the Z-ring

3. In Figure 3, a legend for the colors used to indicate residues identified by various methods should be included within the figure to increase clarity.

4. The phrase “This cavity is ~15 Å wide by ~30 Å tall and is large enough to accommodate a lipid II molecule” is taken directly from another paper and should be re-written.

5. Typo on line 370 – “shown up” should be “show up”

6. The AlphaFold2 model is described as a “homology model” which isn’t quite correct. AlphaFold2 incorporates data from evolutionary sequence analysis, protein folding properties learned from the PDB, and homologous structures, resulting in models that are different from (and more accurate than) conventional homology models.

Reviewer #2: This paper from Li an co-workers investigates the function of the SEDS protein FtsW responsible for building the peptidoglycan (PG) cell wall at the division site. In the last several years, it has become clear that complexes formed by PG polymerases of the SEDS family and class B PBPs (bPBPs) that crosslink PG are the essential PG synthases of the elongation and division machineries of bacteria. The SEDS protein RodA of the elongation machinery has been characterized structurally and subjected to mutagenesis to identify residues required for its activity. As a result, regions of the protein thought to form the active site have been identified.

In this paper, the authors use a clever genetic screen to identity dominant-negative mutants of FtsW, the cell division counterpart of RodA. When mapped onto a model of the FtsW structure, the altered residues in the isolated mutants cluster around what looks to be the active site pocket of FtsW that is analogous to that previously described for RodA. Cell biological and biochemical studies of the altered proteins support the conclusion that these altered FtsW proteins are indeed defective for catalysis.

Overall, the paper is well written, and the results are clearly presented. I also agree with the conclusions. However, the paper could be improved by the addition of some quantitation of the protein localization experiments and the addition of some controls for the biochemical assays.

Major Points:

1) The biochemical experiments in Figure 9 lacks a control to show that FtsI(PBP3) purification is specific for cell producing tagged FtsW or its derivatives. At a minimum, I would like to see an untagged FtsW control showing that the proteins isolated via NiNTA chromatography require the tag. Even better would be the addition of control FtsW mutants with amino acid changes that are known to disrupt the interaction with PBP3 (or alternatively PBP3 mutants known to be defective in interacting with FtsW).

2) As in Figure 9, the biochemical experiments in Figure 10 lack a control for the specificity of the observed signal. Lipid II is hydrophobic and has the propensity to non-specifically associate with other hydrophobic entities, such as membrane proteins in detergent. Therefore, the Lipid II binding experiments require a controls - detergent with no protein added, and detergent micells with a control polytopic membrane protein not expected to bind lipid II. Without these controls, the anisotropy data is difficult to interpret and could be the result of non-specific interactions with lipid II.

3) The micrographs characterizing localization of the FtsW mutants and other divisome proteins or septal PG synthesis when these mutants are produced require some quantification in addition to the micrographs shown. For example, the number of rings observed per cell length should be reported for each experiment, or some equivalent measurement to compare the different effects of the mutants on divisome formation and activity.

**Have all data underlying the figures and results presented in the manuscript been provided?**

Reviewer #1: Yes

Reviewer #2: Yes

PLOS authors have the option to publish the peer review history of their article (what does this mean?). If published, this will include your full peer review and any attached files.

Reviewer #1: No

Reviewer #2: No

---

## [Editor Report · Decision Letter 1]

14 Dec 2021

Dear Dr Du,

We are pleased to inform you that your manuscript entitled "Identification of the potential active site of the septal peptidoglycan polymerase FtsW" has been editorially accepted for publication in PLOS Genetics. Congratulations!

Yours sincerely,

Daniel B. Kearns

Associate Editor

PLOS Genetics

Lotte Søgaard-Andersen

Section Editor: Prokaryotic Genetics

PLOS Genetics

Comments from the reviewers (if applicable):

**Data Deposition**

http://datadryad.org/submit?journalID=pgenetics&manu=PGENETICS-D-21-01295R1

**Press Queries**

---

## [Editor Report · Acceptance letter]

29 Dec 2021

PGENETICS-D-21-01295R1 

Identification of the potential active site of the septal peptidoglycan polymerase FtsW 

Dear Dr Du, 

We are pleased to inform you that your manuscript entitled "Identification of the potential active site of the septal peptidoglycan polymerase FtsW" has been formally accepted for publication in PLOS Genetics! Your manuscript is now with our production department and you will be notified of the publication date in due course.

With kind regards,

Zsofia Freund

PLOS Genetics

On behalf of:
